# A Longitudinal Exploration of Match Running Performance during a Football Match in the Spanish La Liga: A Four-Season Study [note 1]

**DOI:** 10.3390/ijerph18031133

**Published:** 2021-01-28

**Authors:** Eduard Pons, José Carlos Ponce-Bordón, Jesús Díaz-García, Roberto López del Campo, Ricardo Resta, Xavier Peirau, Tomas García-Calvo

**Affiliations:** 1Sports Performance Area, FC Barcelona, 08028 Barcelona, Spain; edu.pons.a@gmail.com; 2Faculty of Sport Sciences, University of Extremadura, 10003 Cáceres, Spain; jdiaz@unex.es (J.D.-G.); tgarciac@unex.es (T.G.-C.); 3LaLiga Sport Research Section, 28043 Madrid, Spain; rlopez@laliga.es (R.L.d.C.); rresta@laliga.es (R.R.); 4National Institute of Physical Education of Catalunya, 25192 Lleida, Spain; xpeirau@gencat.cat

**Keywords:** longitudinal study, match running performance, professional soccer leagues, sports performance, external load

## Abstract

This study aimed to analyze and compare the match running performance during official matches across four seasons (2015/2016–2018/2019) in the top two professional leagues of Spanish football. Match running performance data were collected from all matches in the First Spanish Division (Santander; *n* = 1520) and Second Spanish Division (Smartbank; *n* = 1848), using the Mediacoach^®^ System. Total distance and distances of 14–21 km·h^−1^, 21–24 km·h^−1^, and more than 24 km·h^−1^, and the number of sprints between 21 and 24 km·h^−1^ and more than 24 km·h^−1^ were analyzed. The results showed higher total distances in the First Spanish Division than in the Second Spanish Division (*p* < 0.001) in all the variables analyzed. Regarding the evolution of both leagues, physical demands decreased more in the First Spanish Division than in the Second Spanish Division. The results showed a decrease in total distance and an increase in the high-intensity distances and number of sprints performed, although a clearer trend is perceived in the First Spanish Division (*p* < 0.001; *p* < 0.01, respectively). Knowledge about the evolution of match running performance allows practitioners to manage the training load according to the competition demands to improve players’ performances and reduce the injury rate.

## 1. Introduction

The external load of soccer matches has been studied in depth over the last two decades, which has improved knowledge on its evolution and trends [1]. Thus, different variables have been analyzed, usually related to the distance covered by the players at different intensities [2], and it should be noted that soccer match running performance has evolved, with significant increases in high-intensity actions [3]. Match physical demands can vary depending on the tactical planning, the opposite team’s playing style or the tactical–technical demands [4]. Research has also shown that these changes could be related to differences between soccer leagues [5]. However, to the best of our knowledge, there are no updated studies on how efforts have evolved in professional leagues’ full seasons. In addition, we found no studies of the analysis and comparison of match running performance from several seasons between two professional soccer leagues to update our knowledge about physical differences at the competitive level and in the evolution of football.

Regarding the comparison of match running performance between professional soccer leagues, a previous study analyzed the external load of the top three leagues in English soccer: the FA Premier League, Championship, and League One [5]. This study concluded that the players in the Premier League, compared to players in the lower leagues such as the Championship and League One, covered less total distance and had fewer high-intensity running distances (*p* < 0.01). A related study collected physical demand data over four seasons (2006–2010) in two top leagues of English soccer, with similar external load data [6]. Players of the Championship League (2nd) covered more total distance than players of the Premier League (1st). In addition, Championship players covered more high-intensity running distance and performed more sprinting-intensity actions than Premiership players. However, recent research has found the opposite results in this area of study. In this way, authors described and compared the match running performance of the teams of the Spanish First and Second Division leagues during the 2015–2016 season, showing that the Spanish First Division teams covered more total distance than the Spanish Second Division teams [7]. There were differences in the distance covered at high intensity and very high intensity, where teams from the First Division covered more meters at these intensities. In this line, similar results were reported in the analysis of the match running performance of three professional soccer leagues in Norwegian football [8]. They found a higher total distance in the Norwegian first league teams, but differences were nonsignificant. Concerning high-intensity running distances, Norway’s first league teams covered higher sprinting distances than Norway’s second and fourth league teams (*p* < 0.05). Thus, the most recent studies agree on the presence of higher match physical demands (total and at high intensity) in the top professional soccer leagues.

On the other hand, research of the evolution of external load has shown that total distances have been stable over the period from 1967 to 2012 [9]. However, it has also demonstrated that total distance has increased by 2% in the English Premier League over seven consecutive seasons (2006/2007–2012/2013), whereas high-intensity running and sprint distances have increased by 30–50% [3]. Moreover, a longitudinal study of the World Cup final soccer games reported that the soccer game trend evolved towards shorter, higher intensity play periods because players covered a higher sprint distance and they performed sprints more frequently [10]. Although the evolution of match running performance has also been analyzed by ranking tiers, similar trends have been found for all tiers. In this sense, one study reported that, during seven consecutive seasons in the English Premier League, there was an increase in high-intensity running distance (40%) and leading (15%) and explosive (25%) sprints for all tiers, although the average distance covered per sprint decreased [11]. Thus, changes have been observed in the external load of soccer competitions over the last few years. It is difficult to attribute these findings to a single factor. These changes could be explained through the increases in the competition levels of the leagues, the evolution of movement patterns, training specificity based on match physical demand data or a new approach to training [12]. It also could be related to the playing formation or, possibly, the recruitment of players with more explosive characteristics [1,7,11,13].

There are few studies on the evolution of external load over several years. Most of them are outdated and only analyzed the English Premier League. In addition, even if some works compare leagues or analyze the evolution of external load, there are no studies comparing the evolution of leagues of different levels over several years. Therefore, the aim of this study was to analyze and compare the evolution of match running performance between LaLiga Santander (LL1) and LaLiga Smartbank (LL2) across four seasons (2015/2016–2018/2019).

Based on the aforementioned studies [3,7,11], the authors established the following hypotheses. Concerning the match running performance comparison, we expected that the total distances, the distances covered at high intensity, and the number of very high-intensity running efforts would be higher in LL1 than in LL2.

On the other hand, we expected that the total distance, the distances covered at high intensity, and the number of very high-intensity running efforts would increase in both professional soccer leagues across the four seasons analyzed.

## 2. Materials and Methods

### 2.1. Participants

The sample included observations of all the matches played over four seasons in LL1 and LL2 (2015/2016, 2016/2017, 2017/2018, and 2018/2019). Two observations were made by match, and one by team. In LL1, 752 team match observations were included in the 2015/2016 season; 744 team match observations were included in the 2016/2017 season; 723 observations were included in the 2017/2018 season and, finally, 731 observations were included in the 2018/2019 season. Similarly, in LL2, 700 team match observations were included in the 2015/2016 season; 744 team match observations were included in the 2016/2017 season; 870 observations were included in the 2017/2018 season and, finally, 731 observations were included in the 2018/2019 season. In addition, 784 observations were excluded due to technical problems in the data collecting system or adverse weather conditions during the match, leading to a total of 5952 team match observations.

### 2.2. Design and Procedures

Match running performance data were collected by a multicamera tracking system called Mediacoach^®^. This system assesses the distance covered in meters by teams and the number of high-intensity sprints (LaLiga™, Madrid, Spain). It consists of a series of super 4K-High Dynamic Range cameras based on a positioning system (Tracab—ChyronHego VTS) that records and analyzes X and Y positions for each player from several angles, thus providing real-time three-dimensional tracking (tracking data are recorded at 25 Hz). Mediacoach^®^ has been proven to be both reliable and valid and has been used in previous studies [14,15,16]. Data were provided to the authors by LaLiga^TM^, and the study received ethical approval from the University of Extremadura, Vice-Rectorate of Research, Transfer and Innovation—Delegation of the Bioethics and Biosafety Commission (Protocol number: 153/2017).

### 2.3. Study Variables

Similarly to previous studies [17,18,19], the physical demand variables were recorded for each match: (1) total distance covered by soccer teams in meters (TD); (2) distance covered between 14 and 21 km·h^−1^ (i.e., High-Intensity Running Distance = HIRD); (3) distance covered between 21 and 24 km·h^−1^ (i.e., Very High-Intensity Running Distance = VHIRD); (4) distance covered at more than 24 km·h^−1^ (i.e., Sprinting Distance = SpD). These variables were shown and analyzed by matches and separated by halves (first and second half). In addition, the number of sprints performed was registered, as well as (5) the number of very high-intensity running sprints at 21–24 km·h^−1^ (i.e., SpVHIR), and (6) the number of sprints at more than 24 km·h^−1^ (i.e., SP). All efforts that implied a minimum movement of one meter, which was maintained for a 1 s minimum, were recorded. Any recording at a speed of over 80% of the value of that category (i.e., >24 km·h^−1^) was considered as a single register. All these variables show total team values (i.e., all players who participated in matches, starters, nonstarters and substitutes).

### 2.4. Data Analysis

The statistical program SPSS 25.0 was used (Armonk, NY: IBM Corp, 2017) to analyze and treat the data. Firstly, a two-way Analysis of Variance (ANOVA) was used to explore the main differences between the two professional soccer leagues for external load variables (i.e., variables related to distances covered and the number of sprints) across matches and halves. Subsequently, a 2 × 4 Multivariate Analysis of Variance (MANOVA) was used to examine the differences between the two professional soccer leagues across four seasons in different subsets of dependent variables. A split file, where data were separated by seasons, was used to carry out a posthoc comparison between the professional soccer leagues, using Bonferroni posthoc analyses. Thus, MANOVA investigated the evolution of the external load variables, where season and league (LL1 or LL2) were independent variables. Statistical significance was set at *p* < 0.05, *p* < 0.01, and *p* < 0.001.

## 3. Results

Table 1 shows the mean match running performance comparison between LL1 and LL2 across the four league seasons. We observed a higher TD in LL1 than in LL2 (*p* < 0.001). In the analysis of TD by halves, in LL1, TD decreased over the match, as TD was higher in the first half than in the second half, whereas this trend was the opposite in LL2. Similarly, HIRD was higher in LL1 than in LL2 (*p* < 0.001). Concerning the analysis of the HIRD by halves, this variable was higher in the first half than in the second half in both leagues. VHIRD and SpD were also higher in LL1 than in LL2 (*p* < 0.001). These two variables were higher in the second halves for these two leagues. Finally, SpVHIR and SP were higher in LL1 (*p* < 0.001).

Table 2 shows the evolution of TD and HIRD in LL1 and LL2 over these four seasons. We can observe a progressive decrease in TD, especially in LL1. Furthermore, during the second half, TD decreased more in LL2 than in LL1, where it remained more stable. HIRD showed a slight increase in LL1 and a slight decrease in LL2. Concretely, during the first half, HIRD increased slightly in both professional soccer leagues over the four seasons. However, during the second half, HIRD increased in LL1, whereas in LL2, there was a decrease.

The main difference in the evolution of these professional soccer leagues was the distance covered at very high intensity and sprinting, as shown in Table 3. VHIRD and SpD increased across these four seasons, especially in LL1 (*p* < 0.001 and *p* < 0.001, respectively). During the first half, VHIRD increased significantly in both leagues. Likewise, VHIRD also increased during the second half over the four seasons and in LL1 this increase was significant. In addition, VHIRD was higher in LL1 than in LL2 (*p* < 0.001). For SpD, significant increases were obtained in both leagues (*p* < 0.001). Concretely, in both halves, SpD increased significantly over the four seasons, but it was higher in LL1 than in LL2 (*p* < 0.01).

Finally, Table 4 shows the evolution of SpVHIR and SP across the four seasons and the comparison between the two professional soccer leagues. For SpVHIR, significant increases were found in both leagues (*p* < 0.001). Moreover, SpVHIR was higher in LL1 than in LL2 (*p* < 0.001). On the other hand, SP increased over the four seasons and, in LL2, this increase was significant. SP was also higher in LL1 (*p* < 0.001) than in LL2.

## 4. Discussion

This study aimed to analyze and compare the evolution of the match running performance between the top two professional Spanish leagues (LL1 and LL2) across four seasons: 2015/2016–2018/2019. The main findings of the study showed that TD, VHIRD, and SpD were higher in LL1 than in LL2. Concerning the comparison between the first and second halves, we found that high-intensity efforts increased in the second half, especially in LL1. The match running performance evolved during these seasons, showing different changes between the two leagues. Specifically, TD decreased significantly in LL1, whereas VHIRD and SpD increased progressively in both leagues. SpVHIR also increased significantly in both leagues, whereas SP increased significantly only in LL2.

Firstly, concerning the match running performance comparison, we expected that all the physical variables analyzed in the present study would be higher in LL1 than in LL2. The results showed that external load was higher in LL1 than in LL2. In particular, the distances covered at high intensity and the number of high-intensity efforts were significantly higher in LL1. These results showed that the league at the higher competitive level had higher physical demands during matches. Our findings agree with previous studies [7,8,20], which compared the top two Spanish and Norwegian professional soccer leagues, finding that the top-tiered leagues were more physically demanding. Several explanations could be used to interpret our results. One reason could be the physical capacity of the players of these teams, such that the LL1 clubs contributed to improving the match running performance of their players [11]. Another reason could be related to the playing formation used by LL1 teams, as certain playing formations imply higher external loads, and LL1 teams may use these more demanding playing formations [13]. Concerning the differences between halves of the matches, it can be observed that the first half of LL1 is more demanding than the second half.

Secondly, with respect to the evolution of match running performance during these four seasons, we expected an increase in total distance, the distances covered at high intensity, and the number of very high-intensity running efforts. On the contrary, the changes showed significant decreases in TD in both professional soccer leagues. A possible cause of this may be the playing style used by teams of LaLiga [21] because, in recent years, there has been a gradual increase in teams that prioritized ball possession, confirming that in ball control plays with few transitions players covered less total running distances, although greater distances were covered at high intensity [22]. In addition, the introduction of Video Assistant Referee (VAR) has led to a decrease in effective game time, which has contributed to the decrease in TD [23,24].

In agreement with our hypothesis, where we expected an increase in the distances covered at high intensity and the number of very high-intensity running efforts, the results showed that significant increases in distances covered and efforts performed at high intensity were obtained during the four seasons. In this sense, the significant increases in HIRD and VHIRD are indicators of the evolution and changes occurring in soccer, where players are now trained to perform more high-intensity actions. This has probably been caused by the current training perspective, which increases the presence of high-intensity stimuli according to the competition demands and it decreases the rate of injuries, as achieving optimal player performance while minimizing the risk of injury is the main objective [12,25,26]. These types of efforts are keys to achieving high performances in soccer [27,28] and they are important in decisive situations in professional football. They are the most dominant actions when scoring goals [29]. In this sense, in the 2018/2019 season, VAR was added, which promoted longer recovery times, where high-intensity efforts predominate [24]. Another possible reason could be the tactical evolution of football. Today’s models and playstyles tend to advance defensive pressure lines, resulting in larger spaces and more actions performed at high intensity to take advantage of these spaces.

When examining the match running performance separated by halves, TD decreased in LL1 across the second half, contrary to the results shown in LL2, where TD increased. In addition, in LL1, the decrease in TD in the second half was less than in the first halves of the matches. These results could be explained by the high equality between the teams in LL1 and LL2, where the matches are usually decided in the second half. The decrease in TD in LL1 is further supported by the fact that LL1 teams performed a large number of high-intensity efforts compared to LL2 during the first half, which could cause a decrease in TD during the second half [30].

Finally, concerning the comparison of the evolution between the two professional soccer leagues, we found that in LL1 there is a trend toward a progressive increase in VHIRD and SpD, especially in the second half, whereas in LL2, the trend is not clear. On the other hand, VHIRD and SpD increased during the second halves in both professional soccer leagues, contrary to the results reported in previous studies [31]. A possible reason for these results is the higher TD and high-intensity efforts performed by the substitutes during the second halves [32]. Although we stated that the equality between teams was higher in LL2, another possible explanation is the increase in the effect of match status during the second halves. In both leagues, time pressure is higher in the second half. For example, it is not the same to be losing 1–0 at half-time as at 80 min. The effects of time pressure and match status probably increase high-intensity actions [17,33].

### 4.1. Limitations and Future Perspectives

Taking into account the characteristics of the present study and the novelty of this topic, we considered some limitations with a view to future research. In the 2018/2019 season, VAR was added, which has promoted longer recovery times. In future investigations, we should analyze the differences in the external load before and after the implementation of VAR. In addition, we did not analyze other physical variables such as accelerations and decelerations, which are part of the external load of soccer matches [34]. Thus, these types of physical variables must be analyzed to obtain more information about the match running performance of the competition. Finally, another possible study would be about the different evolution of each team across these seasons (e.g., according to classification or playing style).

### 4.2. Practical Applications

Based on the results obtained, some practical applications can be extracted. Firstly, the paradigm of match running performance has changed across the seasons. Thus, it is also necessary for physical training in soccer to evolve in keeping with current match physical demands to optimize the training process. In this sense, knowledge about the match running performance allows coaches to design soccer training with the correct stimuli to optimize players’ performances. In this regard, this type of stimuli constitutes a methodology for injury prevention and could reduce the injury rate of soccer players. In addition, the evolution of high-intensity efforts is very important in designing specific training tasks that reproduce competition demands. Finally, it was found that the Spanish LL1 is more demanding than LL2, and this information is very important to practitioners who are training in each professional soccer league, since it allows them to discern the different external loads in both the first and second divisions.

## 5. Conclusions

The present research describes and compares the differences in match running performances between the top two Spanish professional soccer leagues across four seasons. Firstly, the results showed higher external loads in LL1 than in LL2. Concretely, the distances covered at high intensity are higher in LL1 than in LL2. Secondly, the decrease in total distance and the increase in distance covered and efforts performed at high intensity are the main changes in the external load of soccer in both leagues. Finally, VHIRD and SpD increased during the second halves in both professional soccer leagues. In summary, we must take into account the evolution of the match running performance in training and the teams’ playing styles to ensure that players are trained to perform more high-intensity efforts during the matches.

## Figures and Tables

**Table 1 ijerph-18-01133-t001:** Differences between both professional soccer leagues in match running performance.

	LL1		LL2		F	*p*
	M (%)	SD	M (%)	SD		
TD (m)	109,135	4355	107,895	4110	126	0.00 (***)
TD 1st Half (m)	54,826 (50.24%)	2390	53,935 (49.99%)	2386	205	0.00 (***)
TD 2nd Half (m)	54,309 (49.76%)	2664	53,960 (50.01%)	2570	26	0.00 (***)
HIRD 14–21 km·h^−1^ (m)	22,436 (20.56%)	2182	21,727 (20.14%)	2005	169	0.00 (***)
HIRD 1st Half (m)	11,395 (10.44%)	1222	10,971 (10.17%)	1129	191	0.00 (***)
HIRD 2nd Half (m)	11,041 (10.12%)	1186	10,756 (9.97%)	1129	89	0.00 (***)
VHIRD 21–24 km·h^−1^ (m)	3019 (2.77%)	385	2838 (2.63%)	378	331	0.00 (***)
VHIRD 1st Half (m)	1504 (1.38%)	230	1409 (1.31%)	223	255	0.00 (***)
VHIRD 2nd Half (m)	1515 (1.39%)	234	1429 (1.32%)	231	202	0.00 (***)
SpD > 24 km·h^−1^ (m)	2905 (2.66%)	490	2687 (2.49%)	481	296	0.00 (***)
SpD 1st Half (m)	1437 (1.32%)	291	1329 (1.23%)	279	209	0.00 (***)
SpD 2nd Half (m)	1467 (1.34%)	304	1357 (1.26%)	299	196	0.00 (***)
SpVHIR 21–24 km·h^−1^	264 (62.12%)	30	249 (62.41%)	30	354	0.00 (***)
SP > 24 km·h^−1^	161 (37.88%)	23	150 (37.59%)	22	287	0.00 (***)

Note: *** *p* < 0.001; TD = Total distance, HIRD = High-intensity running distances, VHIRD = Very high-intensity running distances, SpD = Sprinting distance, SpVHIR = Sprints at very high-intensity running, and SP = Sprints at more than 24 km/h; LL1: LaLiga Santander; LL2: LaLiga Smartbank; % = percentage of the total distance covered. The percentage of SpVHIR and SP takes into account the sum of both variables.

**Table 2 ijerph-18-01133-t002:** Multivariate Analysis of Variance (MANOVA) to compare TD and HIRD between seasons and professional soccer leagues.

		LL1		*p*	LL2		*p*	F	Sig.	Eta	Power
Variables	Season	M	SD		M	SD					
TD (m)	15/16	109,368	4376	^d^	108,176	3973	^bd^	1.53	0.20	0.001	0.41
	16/17	109,241	4319	^d^	107,581	4082	^ac^				
	17/18	109,321	4189	^d^	108,205	4238	^bd^				
	18/19	108,603	4495	^abc^	107,530	4062	^ac^				
TD 1st Half (m)	15/16	55,009	2387	^d^	53,974	2244		3.04	0.03	0.002	0.72
	16/17	54,900	2381	^d^	53,775	2230	^c^				
	17/18	54,861	2395	^d^	54,206	2520	^bd^				
	18/19	54,526	2374	^abc^	53,707	2500	^c^				
TD 2nd Half (m)	15/16	54,358	2660		54,201	2609	^bd^	1.67	0.17	0.001	0.44
	16/17	54,340	2604		53,806	2618	^a^				
	17/18	54,460	2546	^d^	53,999	2578					
	18/19	54,077	2827	^c^	53,822	2434	^a^				
HIRD 14–21 km·h^−1^ (m)	15/16	22,304	2050	^c^	21,743	1987	^bc^	1.68	0.17	.001	0.44
	16/17	22,267	2112	^c^	21,383	2022	^acd^				
	17/18	22,709	2322	^ab^	22,044	2023	^abd^				
	18/19	22,472	2217		21,688	1910	^bc^				
HIRD 1st Half (m)	15/16	11,335	1167	^c^	10,922	1103	^c^	1.50	0.21	0.001	0.40
	16/17	11,307	1189	^c^	10,810	1123	^c^				
	17/18	11,515	1293	^ab^	11,174	1154	^abd^				
	18/19	11,427	1230		10,939	1087	^c^				
HIRD 2nd Half (m)	15/16	10,969	1135	^c^	10,821	1160	^b^	2.85	0.04	0.001	0.69
	16/17	10,960	1140	^c^	10,573	1142	^acd^				
	17/18	11,194	1259	^ab^	10,869	1120	^b^				
	18/19	11,044	119		10,749	1062	^b^				

Note: TD = total distance and HIRD = high-intensity running distances; LL1: LaLiga Santander; LL2: LaLiga Smartbank. Posthoc comparisons: a = significant differences compared with 2015/2016 season; b = significant differences compared with 2016/2017 season; c = significant differences compared with 2017/2018 season; d = significant differences compared with 2018/2019 season.

**Table 3 ijerph-18-01133-t003:** MANOVA to compare VHIRD and SpD between seasons and professional soccer leagues.

		LL1		*p*	LL2		*p*	F	Sig.	Eta	Power
Variables	Season	M	SD		M	SD					
VHIRD 21–24 km·h^−1^ (m)	2015/2016	3020	375		2817	47	^c^	7.19	0.00	0.004	0.98
	2016/2017	2988	385	^d^	2782	47	^cd^				
	2017/2018	3013	384		2907	49	^abd^				
	2018/2019	3056	396	^b^	2836	50	^bc^				
VHIRD 1st Half (m)	2015/2016	1515	233		1392	358	^c^	9.65	0.00	0.004	0.99
	2016/2017	1485	230	^d^	1383	362	^c^				
	2017/2018	1492	222		1448	392	^abd^				
	2018/2019	1523	235	^b^	1406	385	^c^				
VHIRD 2nd Half (m)	2015/2016	1504	225	^d^	1424	210	^c^	2.86	0.04	0.005	0.69
	2016/2017	1503	225		1399	217	^c^				
	2017/2018	1521	244		1459	232	^ab^				
	2018/2019	1533	241	^a^	1429	225					
SpD > 24 km·h^−1^ (m)	2015/2016	2873	468	^d^	2630	28	^c^	3.99	0.01	0.003	0.84
	2016/2017	2860	502	^cd^	2636	29	^c^				
	2017/2018	2930	486	^b^	2777	31	^abd^				
	2018/2019	2959	500	^ab^	2689	30	^c^				
SpD 1st Half (m)	2015/2016	1432	286		1299	491	^c^	5.73	0.00	0.002	0.95
	2016/2017	1413	303	^d^	1293	466	^cd^				
	2017/2018	1440	284		1382	477	^abd^				
	2018/2019	1464	289	^b^	1335	475	^bc^				
SpD 2nd Half (m)	2015/2016	1441	285	^cd^	1331	281	^c^	1.56	0.20	0.003	0.41
	2016/2017	1447	295	^cd^	1342	270	^c^				
	2017/2018	1489	316	^ab^	1394	285	^ab^				
	2018/2019	1495	317	^ab^	1354	267					

Note. VHIRD = very high-intensity running distances, SpD = sprinting distance; LL1: LaLiga Santander; LL2: LaLiga Smartbank. Posthoc comparisons: a = significant differences compared with 2015/2016 season; b = significant differences compared with 2016/2017 season; c = significant differences compared with 2017/2018 season; d = significant differences compared with 2018/2019 season.

**Table 4 ijerph-18-01133-t004:** MANOVA to compare number of sprints at different speed levels between seasons and professional soccer leagues.

		LL1		*p*	LL2		*p*	F	Sig.	Eta	Power
Variables	Season	M	SD		M	SD					
No. SpVHIR 21–24 km·h^−1^	2015/2016	263	29	^d^	247	224	^c^	6.85	0.00	0.001	0.98
	2016/2017	262	31	^d^	245	226	^c^				
	2017/2018	264	30		255	241	^abd^				
	2018/2019	268	32	^ab^	249	228	^c^				
No. SP > 24 km·h^−1^	2015/2016	160	22		148	291	^c^	4.93	0.00	0.001	0.91
	2016/2017	160	23		148	300	^c^				
	2017/2018	161	22		155	303	^abd^				
	2018/2019	162	23		150	297	^c^				

Note. SpVHIR = sprints at very high-intensity running and SP = sprints at more than 24 km·h^−1^; LL1: LaLiga Santander; LL2: LaLiga Smartbank. Posthoc comparisons: a = significant differences compared with 2015/2016 season; b = significant differences compared with 2016/2017 season; c = significant differences compared with 2017/2018 season; d = significant differences compared with 2018/2019 season.

## Data Availability

Restrictions apply to the availability of these data. Data was obtained from LaLiga and are available at https://www.laliga.es/en with the permission of LaLiga.

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
