# Peer review of "A Longitudinal Exploration of Match Running Performance during a Football Match in the Spanish La Liga: A Four-Season Study†"

_ijerph, 2021, doi:10.3390/ijerph18031133_

Round 1

Reviewer 1 Report

Title: Evolution of match physical demands on Spanish 3 LaLiga: A 4-season longitudinal study

Major concerns

- Line 124. How was the “number of efforts performed or registered” are being counted as a frequency? Is it when the player ran beyond the threshold speed set for 0.1 second and this is counted as one frequency or does the player need to sustain the threshold speed for a longer duration of time, say 2 seconds before it was counted as performing the effort?  This was not clearly written or explain to the reader in the method section. Please explain clearly the frequency count of the number of efforts. 

- Table 1. This reviewer feels that the comparison between LL1 and LL2 for their running capacities at the various speed should be also done as percentage of their total distance covered rather than only in absolute values. My rationale. Firstly, total distance covered (TD) is not instructive and meaningless. If the players is does more TD, it does not mean that he is doing “useful work” during the match for he might just be running around the pitch aimlessly. We know that work done in the high-intensity running velocities and above are the “key” or is  the real useful running during a match because it has been shown that high-intensity running are the key to creating opportunities to scoring and critical defending. Thus, if a player does a greater amount more high-intensity running and sprints (as a percentage of his TD), then he is doing more “useful” work compared to a player who may in absolute terms ran more high-intensity run and sprint in but when calculate as a percentage of his TD, the value is lower than that of the other player.  By analysing the running velocities of high and sprints as percentages of the TD, one can really visualize whether the player nowadays is doing more useful work as compared to the previous seasons of the league. And similarly, when compared between two leagues, we can really be confident that players in the LL1 is really doing more useful work relative to the player in the LL2 only when we compared the high-intensity running distance and sprints as a percentage of the players TD.  

- Another concern is the issue of players who were substitutes and players who were being substituted during the match. Were these data omitted or were they included in the current data set? The inclusion of the data of the players substitutes and being substituted will impact the overall results of your current data.

- Table 3. Although there were statistical significance differences in the SpVHIR and SP> 24 km·h-1 across the seasons and between LL1 and LL2, my concern is whether these differences are physiologically meaningful was not discussed in the present manuscript. Take for example, the difference in SpVHIR within the LL1 from season 15_16 to season 18_19 is a mere ~30 m over an entire 90 min match. Is this supposedly difference a real ‘evolution’ and is it really physiologically meaningful? What does the differences of 30 m over a match practically translates to in terms of performance of the player during a match – with the additional 30 m can he score more goals or make more passes? The authors should try to provide some practical implications to such differences for observed in the players running distances differences between the 2 leagues and between seasons – if the observed differences are not well explained then this study is only an academic exercise.  

- This reviewer is not convinced of the practical usefulness of this study. There is a need to justify the purpose and aim of the study clearly – authors must implicitly and explicitly explain how does the results of this study can benefits future players/coaches and football as a whole. The study’s conclusion of “paradigm of physical performance has changed across the season”, to this reviewer at least is not really valid or tenable. The term ‘paradigm’ means a substantial or humongous shift – but with the current data indicating changes of distance covered at high speeds of around ~30 m per match, I don’t think this is a huge or substantial evolution of the match from 2015 to 2019. Line 258 to 260, authors mentioned the need to evolve training and design of training programme and also ensure correct stimuli – these are general comments and “motherhood statements” that can be easily formed without much supporting data; therefore i encourage the authors to be bold/direct and should be more specific about their practical implications from their data/results. As an example, with the data/results can the authors reason that training intensity should be increased by 10% now or that players now should do more high-speed running or do more recovery between sprints – what this reviewer is seeking from the authors are more specific, practical and helpful training implications for the players and coaches. If not, this study will merely be an academic exercise!

-  The written English sentences structure require the close attention of an English-native speaker. There are many awkward written sentences and unclear words/terms throughout the manuscript.

Minor issues:

- Title. Author can assume everybody knows about football/soccer; there should be the word “football” somewhere in the title of the manuscript.

- authors used the terms like “high-intensity actions”, “physical demands”, “physical variables”, “physical variables”, “physical performance” etc throughout the match throughout the manuscript seems to imply all physical activities such as running including other movements like tackling, heading, shooting or dribbling (i.e., all football related actions). However, it should be noted that the study only measured running movements and did not measure all other actions. Thus, authors should be careful to state in their sentences what type of activities are being discussed and should be specific in their arguments of the activities, i.e., running alone or all other football actions are being discussed in the sentences.  

- Line 34. Is there such a word “overstudied”? please check the dictionary.

- Eliminate the use of the term “standard” to define your league throughout the manuscript. 

Line 49 to 53. Re-write tis sentence, “This study concluded that the players covered less total distance and fewer high-intensity running distances (p < .01) in the Premier League compared to lower standard leagues, like Championship and League One.” This should be written as “This study concluded that the players in the Premier League as compared to players in the lower leagues, like Championship and League One, covered less total distance and fewer high-intensity running distances (p < .01)”.

- Line 51. What is “performance data”? Is it the technical skills performance during a match or the physical improvements? Must be specific in your discussion. Authors always use generic terms which is not ideal.

- line 53. “with similar results”. Similar results to what? This term is not clear and require a more explanation.

- Line 56 to 60. For the differences between 2 data set that were non-significant, then you cannot say that one is relatively higher or lower than the other. When there is no statistically significant different between two data set then the two data set are the same; you can cannot say there are differences between them.

Line 54 and 55. “championship” and “premiership” should be “Championship” and “Premiership”.

- Line 63. I think you got the reference cited here wrong. It should not be [8] but rather [7]. Please check.

- Line 73 and 73. This sentence is awkward and not clear. I don’t really understand what you trying to say here. Please get someone to re-read your entire manuscript.

- Line 75. What is meant by a “moderate” increased? This is what I meant by term that is not clear because it was not well-defined earlier. 

-  Line 78 to 80. While I do agree that the changes or increased in players running capacities could be due to the reasons mentioned – but there could be other reasons unexplored such as the matches schedule has been reduced or players are much physically fitter now (due to natural selection of the sports). In short, there are various or numerous other potential reasons for the changes and you citing the reference [10] that hamstring injuries have increased annually by 4% to support your point, this reviewer feels, seems to be an odd citation for this point of argument. Please re-write and get a better reference to cite.   

- Line 81. Delete this heading.

- Line 82. Delete “As has been confirmed”.

- Line 83. “most of them old” is not appropriate. I think you are trying to say that “most of them are outdated”.

- Line 84. “object of analysis”. Please use simply term.

- Line 84 and 86. The term “physical demands” here is misleading because it seems to suggest that you are going to include all physical movements like shooting, walking, etc. be specific on what you are measuring which is players’ running capacities /capabilities.  

- Line 89. Based on “the aforementioned studies”. Which studies specifically. Please cite them here.

- Line 90. Delete “Hypothesis (H1)”. This is a manuscript and not a thesis or dissertation format.

- Line 91 . The uninitiated reader will be confused by the terms “high-intensity” and the “high-speed running efforts” which were unfortunately not clearly defined in the introduction section. Thus, the sentence is unclear to the reader. Please define these two terms in your Introduction section clearly first. Ditto for Line 93 to 95.

- Line 96 to 98. I don’t understand the sentence here. You already made a hypothesis between LL1 and LL2 in Line 90 to 92 and then here you wrote you are not making any hypothesis between the two leagues. You are confusing the reader.

- Line 144. “adequate” is not appropriate here. Please re-write the sentence to inform and convince the reader that the measuring tool, i.e., Mediacoach® is both reliable and valid, and please provide clear evidence of this.

- Line 120 and 121 and 122. Delete the term “total” because it will confuse the reader with the variable of “total distance covered”. Just using the term “distance covered between 14-20 km·h-1” is good enough rather than “total distance covered between 14-20 km·h-1.

- Line 120 and others. Use km·h-1 and not km/h. And use the term “between” and not “at”.

- Line 139. “Table 1 shows the comparison of the four seasons mean values for match physical demands between LL1 and LL2” should be written “Table 1 shows the mean match physical demands comparison between LL1 and LL2 across the four league seasons”.  

- All Tables. Please keep the numerical values of your data to either 1 or no decimal point at all. The current 2 decimal points does not add value to your data. In fact it creates messiness to your tables.   

- Line 160, 161, 162. “respect“ is not a scientific term. Use “with respect to” Or “from” is a better word.

- Line  190 and 192. Delete the term “on one hand”.

- Line 197. Should write out the hypotheses and not ask the reader to refer back to hypothesis. You are supposed to make it easy for the reader to read and understand your manuscript and not the other way around.

- Line 202. Delete “first” and used the word “top-tiered”.

- Line 203-205. The reason stating that the players improved their physical skills contributed to improving their physical performance (in this instance physical performance means running performance) does not make sense. How can a player who improves his skill can lead to improvement in his running capability.

- Line 205-207. These 2 sentences are really awkward and makes no sense. “Another reason could be related to the playing formation used by LL1 teams, as these impact physical performance [17]. In addition, it can be observed that the first half of LL1 is more demanding than the second half”. Pease rephrase or re-write. ]”. Your point of argument is incomplete. Need to add more detail or explanation.

- Line 209. Should write out the hypotheses and not ask the reader to refer back to hypothesis 2a, 2b and 2c. You are supposed to make it easy for the reader to read and understand your manuscript and not the other way around.

- Line 211. Delete “Thus“ and replace with “Because”.

 - Line 212. Where is the evidence to suggest that other clubs in LL1 followed the Barcelona playing style?

- Line 213. Replace “fewer total distance covered” with “players covered less total running distances”.

- Line 214 – 215. I thot the time taken to look and analyze the VAR by the referee is not included in the match time, i.e., the match time is stopped and only continue when the VAR is resolved. Hence to me, this it does not affect the effective game time. Therefore, this is an invalid point of argument.

- Line 216. Again, should write out the hypotheses and not ask the reader to refer back to hypothesis 2b and 2c. You are supposed to make it easy for the reader to read and understand your manuscript and not the other way around.

- Line 219. This sentence “Nowadays, players are trained to perform more high-intensity actions”. Where is the evidence or support for this view or argument?

- Line 221. This statement “it means a correct methodology for injury prevention [10]”. Is awkward and not clear. Please re-write.

- Line 230. What do you mean “the decrease of TD in the second half is shown to be more stable than in the first half.” The meaning of the word “stable” is confusing. How can one decrement is stable?

- Line 248. The game has become “more intermittent”. This term does not make any sense. Its either intermittent or continuous. The pattern of match cannot be “more intermittent”. Perhaps you can use term like the “exercise to recovery ratio” during match has increased or decreased because of the VAR.

- Line 262 – 263. This sentence is unclear. Please rephrase.

- Line 273. Authors reasoned that the evolution of the running capacities of the players observed was due to “changes in playing styles” of the clubs over the years. At this juncture, this reviewer is not convinced this is the case. Authors need to provide some evidence of this in the Discussion section before concluding with such a statement.

- Line 328. Reference no 14: No journal name and pages number. Line 332 - same for reference no 15 – no volume and page numbers. Please make sure you ensure consistency in the format your reference.-  sometimes the titles are capitalized and sometimes they are not.

Reviewer 2 Report

Comments and Suggestions attached.

Round 2

Reviewer 1 Report

Major concerns:

  1. The written English in the manuscript is still not good enough. Please get a native English speaker to read your manuscript and improve the sentences structure. As an example, line 24”…. decrease more….”. “more” is not correct English here. Something like “decrease to a greater degree or extent” is more appropriate. Another sentence: “There are few studies of the evolution of external load (of what??) across several years". Another example, line 99, “Two recordings resulting by match….” (this is a funny way of expressing that 2 competing teams’ data were obtained during a single match). There are many more throughout the manuscript which can easily be corrected by a native speaker.

  1. Tables. The comparison between season should be done based on the number of matches played during each of the season. Were equal number of matches played for each of the 4 seasons? If not, then the absolute comparison is not valid because the evolution of the match physical demands could simply be due to the varying number of matches during each of the season. I suggest that the all of the data are reported are done per match for each of the season and for each of the league to allow a more valid and fairer comparison between seasons and between teh 2 leagues, respectively.

  1. Again lack of consistency, the authors used “ external load’, “match physical demands”, “efforts” (line 41 and 286), match running performance (line 273) interchangeable throughout the manuscript. This reviewer suggests to use the term “match running performance” (which is exactly what the authors are measuring) as the single only term use throughout the manuscript.

Title should now be something like: A Longitudinal Exploration of Match Running Performance during a Football Match in the Spanish La Liga: A 4-Season Study

Minor concerns:

- Km per h should be written as km·h-1 . The -1 should be superscript.

- I don’t understand the link between the higher level of intensity in LL1 relative to LL2 with the “teams’ promotion”. 

- the seasons are written either 2018/2019 and at times as 2018_2019. Please be consistent throughout manuscript.

Reviewer 2 Report

I would advise the authors to reconsider the tittle one more time. it is not clearly and smoothly stated. (I mean english language)

good luck
